# Ultrasound-Guided Integrated Musculoskeletal and Vascular Landmark Approach for Access to the Facial Nerve Trunk

**DOI:** 10.3390/life15091396

**Published:** 2025-09-03

**Authors:** Yeui-Seok Seo, Yonghyun Yoon, King Hei Stanley Lam, Sang-Hyun Kim, In-Beom Kim, Kwan-Hyun Youn

**Affiliations:** 1Department of Anatomy, Catholic Institute for Applied Anatomy, College of Medicine, The Catholic University of Korea, Seoul 06591, Republic of Korea; prssys@area88ps.com (Y.-S.S.); amalang@catholic.ac.kr (S.-H.K.); 2Area88 Plastic Surgery Clinic, Seoul 06615, Republic of Korea; 3Department of Orthopaedic Surgery, Gangnam Sacred Heart Hospital, Hallym University College of Medicine, 1 Singil-ro, Yeongdeungpo-gu, Seoul 07441, Republic of Korea; 4Incheon Terminal Orthopedic Surgery Clinic, Inha-ro 489beon-gil, Namdong-gu, Incheon 21574, Republic of Korea; 5International Academy of Regenerative Medicine , Inha-ro 489beon-gil, Namdong-gu, Incheon 21574, Republic of Korea; 6The Board of Clinical Research, The International Association of Musculoskeletal Medicine, Kowloon, Hong Kong; 7MSKUS, 1035 E. Vista Way #128, Vista, CA 92084, USA; 8Faculty of Medicine, The University of Hong Kong, Hong Kong; 9Faculty of Medicine, The Chinese University of Hong Kong, New Territory, Hong Kong; 10Division of Biomedical Art, Incheon Catholic University Graduate School, Incheon 06591, Republic of Korea; artanato@naver.com

**Keywords:** ultrasound-guided injection, facial nerve, hypoglossal nerve, plastic surgery, digastric muscle, neurovascular mapping

## Abstract

**Background:** Ultrasound is increasingly used in plastic surgery for real-time guidance in minimally invasive procedures. However, standardized approaches for targeting the facial nerve (FN) trunk, particularly for motor nerve interventions, remain limited. This study aimed to evaluate the anatomical feasibility of an ultrasound-guided approach to the FN trunk using the posterior belly of the digastric muscle (PBDM) as a landmark. **Methods:** An exploratory feasibility design was used with a single fresh-frozen cadaver to perform ultrasound-guided dye injections targeting the anterior and posterior surfaces of the PBDM. Subsequent layer-by-layer dissection evaluated dye distribution relative to the facial and hypoglossal nerves. Additionally, real-time Doppler ultrasound in a live participant was conducted to visualize adjacent vascular structures, including the occipital and vertebral arteries. **Results:** The FN trunk was located deep to the PBDM and near the stylomastoid foramen. Anterior injections reached the FN trunk, whereas posterior injections followed the trajectory of the hypoglossal nerve. Doppler ultrasound enabled clear visualization of major vascular structures, supporting safe needle trajectory planning. **Conclusions:** This cadaveric feasibility study demonstrates a potentially reproducible ultrasound-guided anatomical approach to the FN trunk using consistent musculoskeletal and vascular landmarks. Incorporating Doppler vascular mapping enhances procedural safety and accuracy, providing a practical framework to facilitate clinical translation of image-guided motor nerve interventions in plastic and reconstructive surgery.

## 1. Introduction

Ultrasound has long been used as both a diagnostic and interventional tool across various medical specialties, including obstetrics, internal medicine, and musculoskeletal care [1]. Recent advances in high-resolution real-time imaging have expanded its applications from peripheral to cranial nerves [1], enabling safer and more precise minimally invasive procedures. In plastic and reconstructive surgery, ultrasound is increasingly employed to enhance procedural safety [2,3], particularly in the facial region for applications such as filler injection guidance [4], vascular mapping [5], and removal of foreign bodies [6].

Despite these advances, the use of ultrasound for motor nerve-targeted interventions, especially involving the facial nerve (FN) trunk, remains underdeveloped [7,8]. Existing studies have primarily focused on superficial muscle layers or detecting asymmetry in cases of facial palsy, with limited relevance to procedures that require precise access to deeper anatomical structures. Moreover, standardized sonographic approaches to the FN trunk—and their integration with vascular mapping—have not been well established. This lack of procedural guidance limits the adoption of ultrasound in FN interventions such as reanimation surgery, decompression, or neuromodulatory injections near the stylomastoid foramen.

Ultrasound-guided hydrodissection has recently emerged as a non-surgical treatment option for neural pathologies [9]. This technique involves the percutaneous injection of a fluid medium to separate a nerve from surrounding fibrotic or compressive tissue. Its primary purposes are mechanical decompression to facilitate nerve gliding and biochemical neuroprotection to modulate the local inflammatory environment. Common injectates include 5% dextrose in water (D5W), known for its potential to reduce reactive oxygen species and neuronal apoptosis with relatively low neurotoxicity, and local anesthetics, often combined with corticosteroids for their anti-inflammatory effects [7,10,11]. Initially applied to sensory nerves [7], it has shown potential in motor nerve applications, including the FN, for conditions such as facial palsy or chronic neuritis [10,11]. However, translating this technique into routine practice requires reproducible, anatomy-based ultrasound protocols that account for both neural and vascular structures.

Therefore, the aim of this exploratory cadaveric study was to define and validate an ultrasound-guided anatomical approach to the FN trunk using consistent musculoskeletal and vascular landmarks. By combining ultrasound-guided dye injection with anatomical dissection and live Doppler vascular imaging, we sought to provide a reproducible framework to enhance the safety and accuracy of motor nerve interventions in the craniofacial region.

## 2. Materials and Methods

### 2.1. Ethical Approval

This study was approved by the Institutional Review Board of The Catholic University of Korea, Seoul St. Mary’s Hospital (IRB No. MC24EISI0041; approval date: 19 April 2024). The cadaveric component was conducted in compliance with institutional anatomical research guidelines. The live Doppler ultrasound component was performed using data from a previously treated patient and qualified for exemption from additional IRB review under institutional and national regulations.

### 2.2. Study Design

An exploratory feasibility design was adopted to evaluate an ultrasound-guided anatomical approach to the FN trunk. One fresh-frozen cadaver was used for ultrasound-guided dye injection, and a live participant was examined for vascular Doppler imaging.

### 2.3. Cadaveric Procedure

A non-embalmed fresh male cadaver (Case ID 25-078) was used for this feasibility study. The subject was 68.8 years old at the time of death, with a height of 166 cm and a body weight of 56 kg. The cause of death was lung cancer, and no prior surgical intervention in the head and neck region was identified. The cadaver was examined within 72 h postmortem, thereby preserving near-physiological tissue elasticity and vascular integrity, which is essential for injection-based anatomical studies.

Ultrasound imaging was first performed on one side of the neck to identify relevant anatomical structures and sonographic landmarks. On the contralateral side, a mixture of red and blue dye was injected, targeting the superior and inferior aspects of the posterior belly of the digastric muscle (PBDM) as visualized on ultrasound. The cadaver was positioned in the lateral decubitus position with induced lateral neck flexion to expose the stylomastoid foramen.

### 2.4. Ultrasound Settings and Injection Technique

Ultrasound was performed using an Alpinion XC90 elite (ALPINION MEDICAL SYSTEMS Co., Ltd., Seoul, Republic of Korea) system with a linear 8 MHz probe (4 cm depth, dynamic range 60), with the focal point aligned to the stylomastoid foramen. Coupling gel was applied to optimize acoustic transmission. Injections were carried out by an orthopedic surgeon (Y.H.Y.) with more than 10 years of musculoskeletal ultrasound experience, using a 23-gauge, 6-cm needle advanced in a posterior-to-anterior direction. At each target site, 2 mL of dyed filler was injected at each target site.

### 2.5. Anatomical Dissection

Dissection was conducted by a professional anatomist with over 10 years of cadaveric dissection experience. A layer-by-layer approach was used to preserve and visualize each branch of the FN. The sequence involved removal of skin and platysma, exposure of the epimysial fascia covering the sternocleidomastoid (SCM) and trapezius muscles, isolation of the greater auricular and lesser occipital nerves, transection and superior reflection of the SCM, and exposure of the PBDM. The distance between the dye and the facial nerve and hypoglossal nerve was measured to assess injection accuracy.

### 2.6. Live Doppler Ultrasound

In the live participant, real-time Doppler ultrasound was performed to visualize adjacent vascular structures, including the occipital and vertebral arteries. The participant was positioned in the same lateral decubitus position with induced lateral neck flexion as the cadaver to ensure anatomical consistency.

## 3. Results

### 3.1. Overall Findings

Our integrated sonographic and anatomical analyses established a reproducible and structured approach for identifying key landmarks and optimizing probe orientation to guide safe and effective access to the FN trunk.

### 3.2. Probe Positioning

We consistently visualized critical anatomical landmarks necessary for FN access using standardized probe manipulation techniques. Common musculoskeletal ultrasonography maneuvers—sliding, fanning, and the heel–toe technique [12]—enabled clear identification of the stylomastoid foramen, mastoid process (MP) [13,14], and PBDM (Figure 1). The probe positioning process was documented in the Appendix A, facilitating procedural reproducibility. The process of image acquisition was recorded in the Video, Supplemental Digital Content provided with this manuscript, which enhances procedural reproducibility and supports clinical application. The ultrasonic probe position is indicated by a blue rectangle in the schematic drawing at the upper right of the ultrasound image.

### 3.3. C1 Detection and Muscles

High-resolution ultrasound allowed reliable visualization of the first cervical vertebra (C1), including its transverse process (C1 TP) and lamina (C1 Lamina) [15]. These bony structures, along with adjacent muscles (the SCM, splenius capitis muscle (SpC), longissimus capitis muscle (LC), and obliquus capitis inferior muscle (OCI)), served as consistent anatomical reference points in both ultrasound imaging and cadaveric dissection (Figure 2 and Figure 3).

Surrounding muscles—including the SCM muscle, splenius capitis muscle (SpC), LC, and OCI—were consistently identifiable on sonographic imaging near the C1 level. Their spatial orientation matched that observed in corresponding anatomical dissection images (Figure 3), confirming their value as reliable anatomical landmarks.

Following anterior translation of the probe, the PBDM [16] structure emerged as a critical muscular landmark for accessing the FN. Its location and morphology were consistent across ultrasound imaging and cadaveric dissection and were distinguishable by a distinct fascial boundary on sonographic imaging (Figure 1 and Figure 4), supporting its feasibility as a real-time procedural guide.

The comparative analysis of ultrasound imaging and anatomical dissection reinforces the reproducibility and structural consistency of C1-based access routes to the FN, thereby supporting the anatomical validity of this landmark-guided technique.

### 3.4. Identification of the Vertebral Artery (VA)

By tilting the ultrasound probe superiorly, the VA was consistently visualized in the upper cervical region. On grayscale imaging (Figure 4), the VA appeared as a hypoechoic tubular structure located deep between the PBDM and the occipital condyle (OC), with the SCM and SpC situated more superficially.

**Figure 4 life-15-01396-f004:**
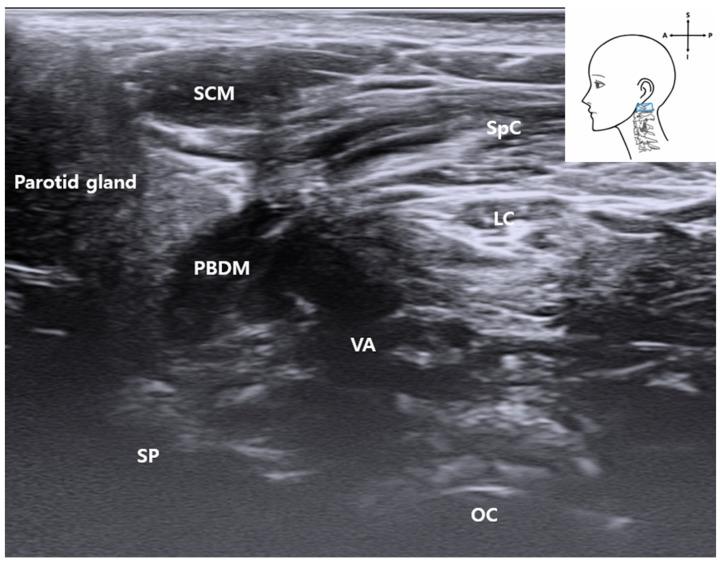
Ultrasound image showing the VA located deep between the posterior belly of the digastric muscle and occipital condyle. The SCM muscle lies superficially. This image was obtained with the probe tilted superiorly using a heel-toe maneuver to visualize the upper cervical region. *SCM*, sternocleidomastoid muscle; *PBDM*, posterior belly of digastric muscle; *SpC*, splenius capitis muscle; *VA*, vertebral artery; *OC*, occipital condyle; *SP*, styloid process; *LC*, Longissimus capitis muscle); blue rectangle, probe position.

Notably, real-time Doppler ultrasound in a living subject confirmed vascular flow in this region (Figure 5), demonstrating the feasibility of identifying the VA during ultrasound-guided procedures. Color flow mapping confirmed arterial dynamics, allowing differentiation from adjacent non-vascular structures.

Anatomically, the VA traverses a narrow corridor between the C1 lamina and the occipital bone, placing it in close proximity with the ultrasound probe when targeting the stylomastoid foramen. This spatial relationship was confirmed by cadaveric dissection and underscores the clinical importance of Doppler screening before needle advancement.

Together, these findings underscore the VA as a critical deep vascular structure that must be identified and avoided during ultrasound-guided interventions targeting the FN. The use of Doppler imaging enhances both procedural safety and anatomical precision.

### 3.5. Identification of the Occipital Artery

Ultrasonographic and anatomical evaluations revealed that the occipital artery (OA) courses posterocranially along the longitudinal axis, located just beneath the PBDM (Figure 6). Cadaveric dissection showed that the OA arises from the external carotid artery (ECA) and runs posteriorly beneath the PBDM, positioned inferior to the posterior auricular artery (PAA). On grayscale ultrasound, the OA appeared as a hypoechoic tubular structure running parallel to and immediately deep to the PBDM in the long-axis view (Figure 7). The occipital bone consistently served as an osseous landmark at the bottom of the image.

In live patient imaging, color Doppler ultrasound confirmed real-time vascular flow in the OA as it coursed obliquely through the scanning field (Figure 8). This enabled clear identification of its trajectory and spatial relationship to adjacent muscular structures.

This anatomical positioning suggests that the OA may intersect the intended needle trajectory during ultrasound-guided interventions near the stylomastoid foramen or hypoglossal nerve, particularly when the approach traverses or lies adjacent to the PBDM. To minimize the risk of vascular injury, pre-procedural Doppler ultrasound is essential for identifying and avoiding the OA.

### 3.6. Visualization of the Stylomastoid Foramen and FN

As the probe was advanced superiorly, the PBDM gradually tapered as it inserted into the MP, revealing the interval between the styloid and MP [17], where the FN emerges superficially.

This region anatomically corresponds to the stylomastoid foramen, although it does not appear as a distinct opening on ultrasound imaging. Similar with other bony foramina of the skull, its identification is indirect, based primarily on characteristic acoustic shadowing rather than a visibly defined aperture (Figure 9). Bone produces high acoustic impedance and typically appears as a hyperechoic linear edge with a corresponding anechoic shadow beneath it. This indirect representation has also been documented in the ultrasound identification of small bony openings, such as the supraorbital notch and infraorbital foramen, where their presence is inferred from disruptions in echo patterns rather than direct visualization [6]. Similarly, the stylomastoid foramen can be inferred sonographically by observing localized acoustic shadowing between the mastoid and SP. This sonographic impression corresponded well with anatomical observations from cadaveric dissection (Figure 9).

### 3.7. Ultrasound-Guided Injection

In the ultrasound-guided injection experiment, the anterior aspect of the PBDM was targeted for needle insertion. Ultrasound imaging confirmed the precise anatomical location, and 2 mL of blue dye was injected at the designated site (Figure 10 and Appendix A: ultrasound-guided filler injection with blue and red dye; injection depth and angle can be found in the video supplement). Subsequent cadaveric dissection demonstrated that the injected dye diffused toward the SP and ultimately reached the area within 2 mm of the FN.

These findings confirm that the selected injection pathway effectively accessed the anatomical course of the FN trunk. In the hypoglossal nerve injection experiment, the posterior aspect of the PBDM was selected as the target site under ultrasound guidance. Red dye was injected into this region to observe its anatomical diffusion pathway (Figure 11; Appendix A: Ultrasound-guided injection).

The hypoglossal nerve itself could not be directly visualized on ultrasound in our study; however, its anatomical location was confirmed through subsequent cadaveric dissection. The nerve lies deep in the cervical region and follows an oblique trajectory relative to the ultrasound beam, rendering it particularly susceptible to anisotropy artifacts and acoustic shadowing. This limitation in sonographic visualization aligns with prior anatomical literature and was anticipated in our imaging protocol [18]. Anatomical verification confirmed that the injected dye distributed along the hypoglossal nerve pathway into the deep fascial plane behind the SP and entered within 2 mm of the hypoglossal nerve. These findings suggest that ultrasound-guided injection via a posterior approach targeting the PBDM offers a feasible anatomical route for accessing the hypoglossal nerve. Moreover, ultrasound-guided injection targeting the PBDM may serve as a reliable and safe anatomical approach for FN and hypogloassal nerve blockade or therapeutic intervention. The PBDM’s location provides a relatively vascular-sparing corridor, reinforcing the feasibility and safety of this technique for clinical application.

## 4. Discussion

This study addresses the absence of standardized ultrasound-guided approaches for accessing the FN trunk. By integrating anatomical dissection with high-resolution and Doppler ultrasound imaging, we established a reproducible technique using the PBDM, C1 TP, and styloid–mastoid region as reliable landmarks. Incorporating vascular mapping of the occipital and vertebral arteries further enhances procedural safety, providing a practical framework for clinical application.

Previous ultrasound-based studies on the FN have primarily focused on superficial branches, omitting the trunk and its deeper anatomical context [2,19]. Most have been limited to cadaveric investigations, lacked vascular correlation, and did not provide clinically applicable procedural pathways. In contrast, our work introduces a deeper, clinically relevant route with concurrent vascular assessment, addressing a critical gap for motor nerve interventions.

Anterior targeting of the PBDM consistently reached the FN trunk, while posterior targeting accessed the hypoglossal nerve. This dual-pathway approach offers flexibility for neuromodulation, nerve blocks, and other interventions. For example, targeting the FN trunk on the affected side can be valuable for managing post-paralytic facial synkinesis or hemifacial spasm with neurolytic agents or for performing diagnostic blocks. Similarly, access to the hypoglossal nerve is relevant for procedures addressing glossopharyngeal neuralgia or tongue dystonia. The vascular-sparing nature of the PBDM corridor minimizes the risk of iatrogenic vessel injury, which is essential for high-precision cranial nerve procedures.

### 4.1. Hydrodissection Potential

Ultrasound-guided hydrodissection, traditionally applied to sensory nerves [7], is emerging as a promising option for motor nerve procedures. The use of 5% dextrose in water (D5W) provides mechanical decompression and biochemical neuroprotection by reducing reactive oxygen species and neuronal apoptosis [10,11], with lower neurotoxicity compared to local anesthetics [7]. Our anatomical findings support the feasibility of D5W-based hydrodissection targeting the FN trunk via the PBDM. While the current cadaveric study used dye to validate the needle trajectory, the approach is directly applicable to hydrodissection procedures in clinical practice, where the injectate would serve both a mechanical and potential therapeutic role.

### 4.2. Limitations

This study has several limitations. First, it was based on a single fresh-frozen cadaver and one live participant, which restricts the generalizability of our findings. Anatomical variation in nerve trajectories, vascular branching patterns, and muscle morphology could significantly influence reproducibility in a broader population. Second, although injection spread was qualitatively assessed, we did not record quantitative outcome measures such as precise diffusion distance, trajectory angles, or depth of needle insertion, which would have strengthened the rigor of our feasibility data. Third, the hypoglossal nerve could not be directly visualized on ultrasound due to anisotropy and acoustic shadowing artifacts, necessitating reliance on post-dissection confirmation. Fourth, procedural factors such as probe angulation, applied pressure, and range of motion were not standardized, which may affect reproducibility between operators. Finally, the feasibility conclusions regarding hydrodissection are preliminary, as the cadaveric injections did not replicate in vivo tissue dynamics, perfusion, or physiologic responses.

Taken together, these limitations underscore that the present findings should be interpreted as preliminary anatomical and sonographic guidance. Larger-scale, multicenter studies with multiple cadavers and clinical participants, incorporating quantitative imaging and procedural standardization, are required to validate the reproducibility, safety, and translational applicability of this technique.

### 4.3. Future Directions

Larger-scale, multicenter studies with quantitative imaging and procedural standardization are needed to validate reproducibility across anatomical variations. Prospective clinical trials should assess safety, efficacy, and functional outcomes in patients undergoing ultrasound-guided interventions for facial and related cranial nerve disorders.

## 5. Conclusions

We present a reproducible ultrasound-guided approach to the FN trunk using the PBDM as a key anatomical landmark. Incorporating musculoskeletal and vascular mapping enhances procedural safety and precision. Further clinical validation is warranted.

## Figures and Tables

**Figure 1 life-15-01396-f001:**
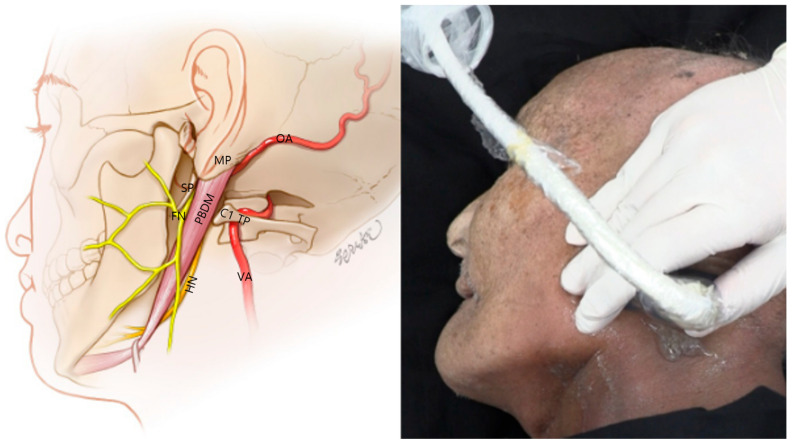
Anatomical schematic (**left**) and cadaveric ultrasound probe positioning (**right**) for ultrasound-guided identification of the facial nerve (FN) trunk. The illustration highlights the FN and hypoglossal nerve (HN) in orange, along with adjacent vascular structures—the vertebral artery (VA) and occipital artery (OA)—in red. The posterior belly of the digastric muscle (PBDM) is shown as a key muscular landmark. Relevant bony landmarks—the styloid process (SP), mastoid process (MP), and C1 transverse process (C1 TP) and lamina—are also depicted to guide accurate localization. The cadaveric image demonstrates real probe placement aligned over these structures, with the probe oriented obliquely to visualize the stylomastoid foramen in ultrasound. SCM, sternocleidomastoid muscle.

**Figure 2 life-15-01396-f002:**
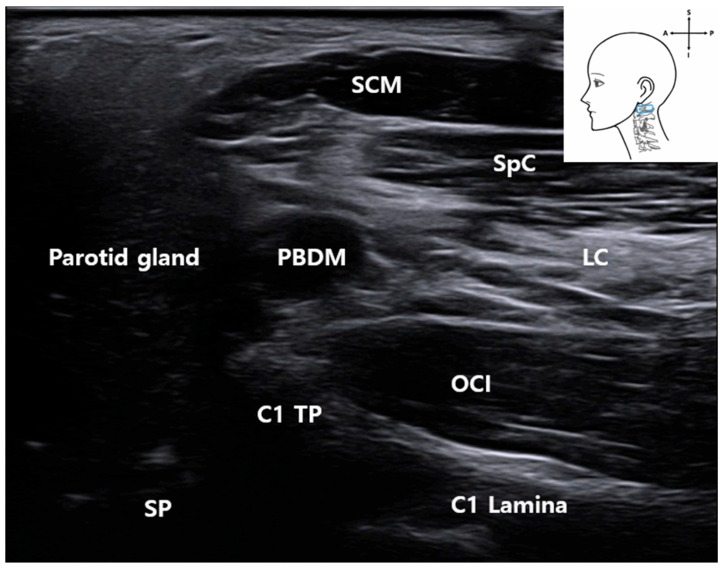
Ultrasound image corresponding to the anatomical dissection of the right lateral neck showing key muscular and bony landmarks related to C1 identification and FN approach. *SCM*, sternocleidomastoid muscle; *SpC*, splenius capitis muscle; *LC*, longissimus capitis muscle; *OCI*, obliquus capitis inferior muscle; *PBDM*, posterior belly of the digastric muscle; *C1 TP*, transverse process of the atlas (C1); *C1 Lamina*, lamina of the atlas (C1); blue rectangle, probe position.

**Figure 3 life-15-01396-f003:**
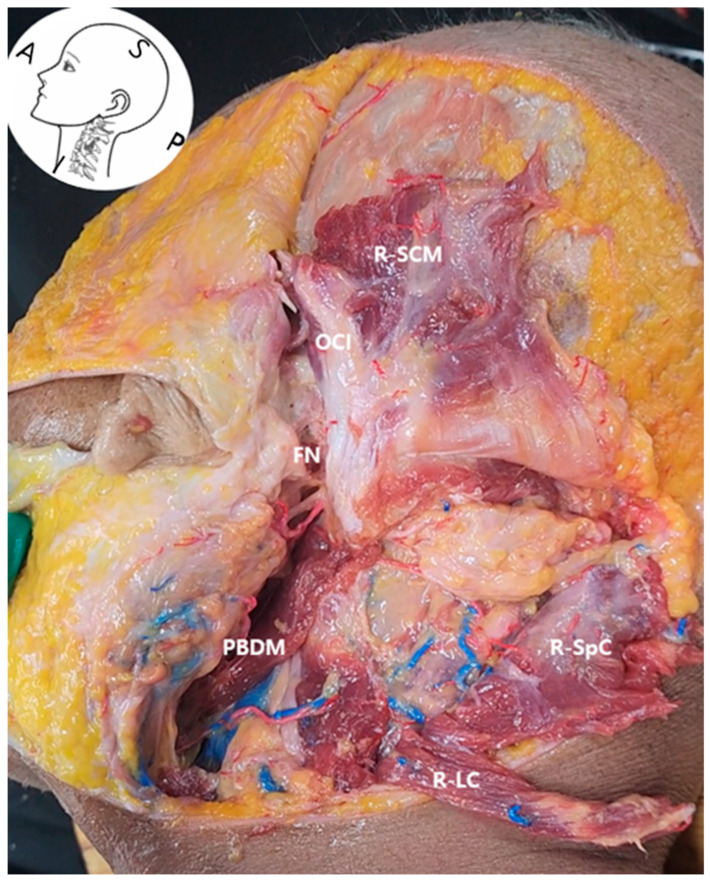
Layer-by-layer dissection of the right lateral neck showing key anatomical structures for ultrasound-guided FN access. *FN*, facial nerve; *PBDM*, posterior belly of the digastric muscle; *R-SCM*, reflected sternocleidomastoid muscle; *R-SpC*, reflected splenius capitis muscle; *R-LC*, reflected longissimus capitis muscle; *OCI*, obliquus capitis inferior muscle.

**Figure 5 life-15-01396-f005:**
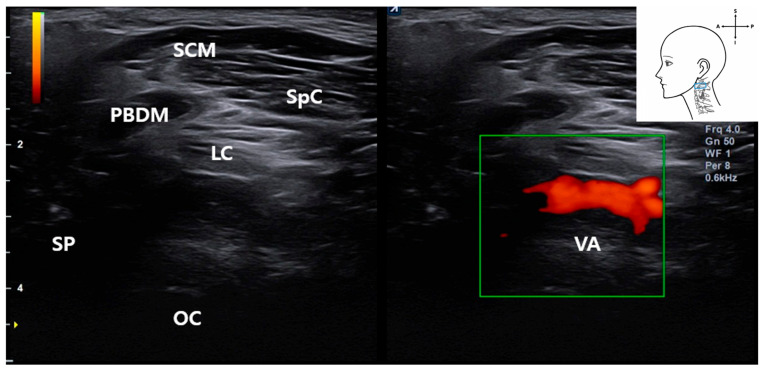
Live dual doppler ultrasound in a living patient confirming visualization of the VA. Real-time vascular flow was detected in the deep cervical plane, suggesting that this structure can be monitored during ultrasound-guided FN interventions. Color flow mapping (green box) confirms arterial characteristics. Blue rectangle, probe position.

**Figure 6 life-15-01396-f006:**
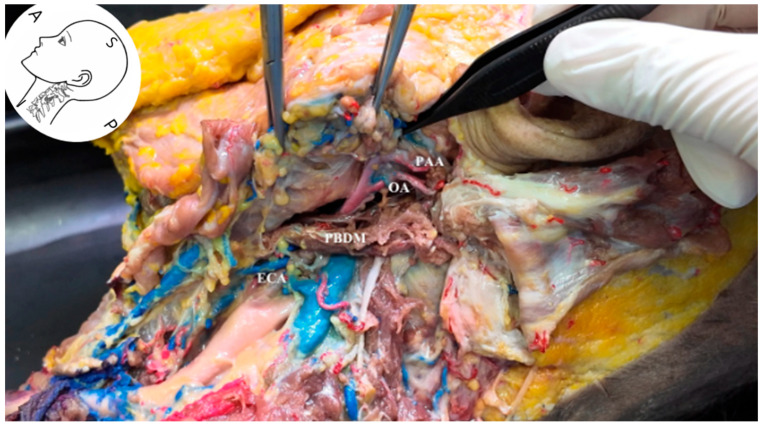
Cadaveric dissection showing the anatomical relationship of the OA. The OA arises from the external carotid artery and courses posterolaterally beneath the posterior belly of the digastric muscle toward the skull base. The PAA is also visible. *ECA*, external carotid artery; *OA*, occipital artery; *PAA*, posterior auricular artery; *PBDM*, posterior belly of digastric muscle.

**Figure 7 life-15-01396-f007:**
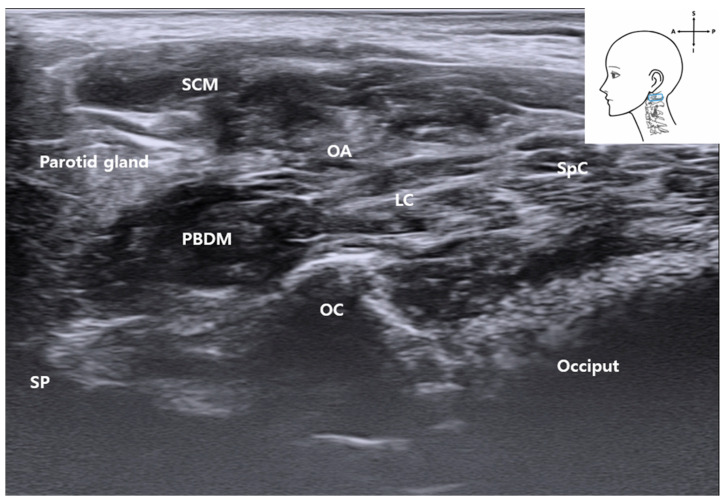
Ultrasound image showing the OA coursing parallel and just deep to the posterior belly of the digastric muscle in a longitudinal (long axis) orientation. The occipital bone (occiput) is seen at the bottom of the image, serving as a deep osseous landmark. *SCM*, sternocleidomastoid muscle; *SpC*, splenius capitis muscle; *LC*, Longissimus capitis muscle; *PBDM*, posterior belly of digastric muscle; *OA*, occipital artery; *OC*, occipital condyle; *SP*, styloid process); blue rectangle, probe position.

**Figure 8 life-15-01396-f008:**
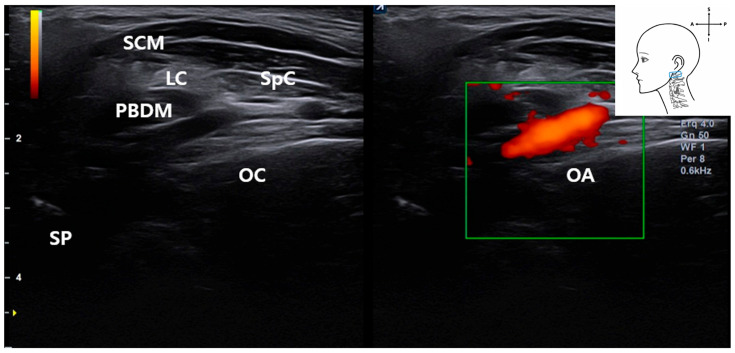
Color Doppler ultrasound image from a live patient showing the OA coursing obliquely through the scanning field. The probe position and Doppler application reveal real-time vascular flow. *SCM*, sternocleidomastoid muscle; *LC*, longus capitis muscle; *SpC*, splenius capitis muscle; *PBDM*, posterior belly of digastric muscle; *OA*, occipital artery; *SP*, styloid process; *OC*, occipital condyle); blue rectangle, probe position.

**Figure 9 life-15-01396-f009:**
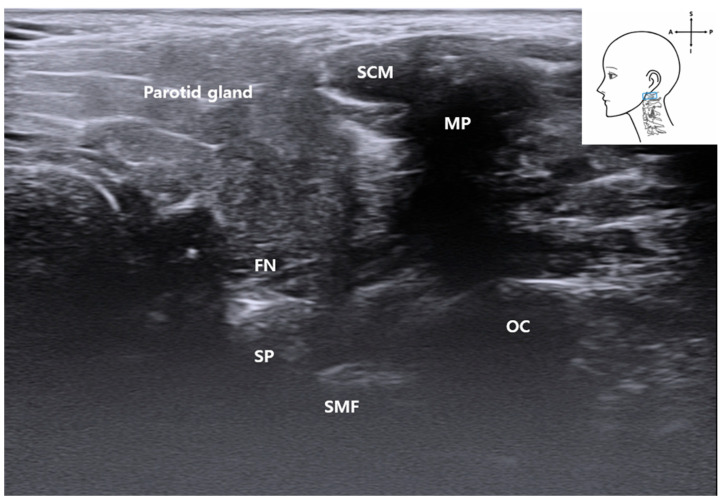
Ultrasound image showing the FN emerging between the mastoid and SP. The stylomastoid foramen is not directly visualized as a discrete opening but inferred from the acoustic shadowing pattern between the bony landmarks. *MP*, mastoid process; *PBDM*, posterior belly of digastric muscle; *FN*, facial nerve; *SP*, styloid process; *OC*, occipital condyle; *SCM*, sternocleidomastoid muscle; *SMF*, stylomatoid foramen); blue rectangle, probe position.

**Figure 10 life-15-01396-f010:**
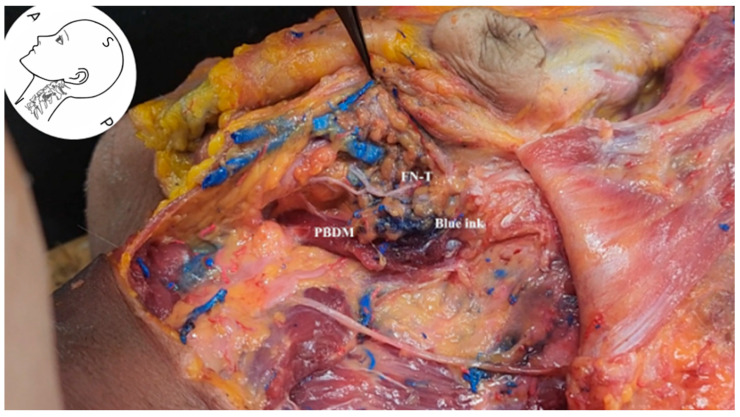
Cadaveric dissection showing the diffusion of blue dye following ultrasound-guided injection targeting the anterior aspect of the posterior belly of the digastric muscle. The dye successfully spread toward the *FN* trunk near the styloid process, confirming the accuracy and feasibility of the ultrasound-based approach. *PBDM*, posterior belly of digastric muscle; *FN-T*, facial nerve trunk; *Blue ink*, diffusion marker.

**Figure 11 life-15-01396-f011:**
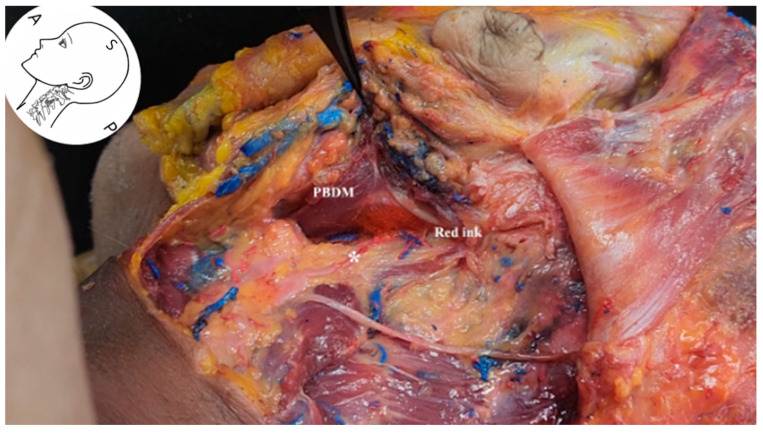
Cadaveric dissection showing red dye distribution following ultrasound-guided injection targeting the posterior aspect of the posterior belly of the digastric muscle. The dye spreads posterior and deep to the styloid process, following the anatomical course of the hypoglossal nerve. This validates the feasibility of ultrasound-guided deep-plane targeting hypoglossal nerve access. *PBDM*, posterior belly of digastric muscle; *Red ink*, diffusion marker; *asterisk*, hypoglossal nerve.

## Data Availability

The original contributions presented in the study are included in the article, further inquiries can be directed to the corresponding author.

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
