# Peer review of "Ultrasound-Guided Integrated Musculoskeletal and Vascular Landmark Approach for Access to the Facial Nerve Trunk"

_life, 2025, doi:10.3390/life15091396_

Round 1
Reviewer 1 Report
Comments and Suggestions for Authors
Dear Authors, I congratulate on this very interesting study on ultrasound-guided approach to the facial nerve trunk. I find the results of your study of great significance to the readership, in spite of the limitations of no having multiple subjects to strengthen the evidence. I recommend to publish.
Author Response
Dear Authors, I congratulate on this very interesting study on ultrasound-guided approach to the facial nerve trunk. I find the results of your study of great significance to the readership, in spite of the limitations of no having multiple subjects to strengthen the evidence. I recommend to publish.
- Thank you for your kind comments and recommendations
Reviewer 2 Report
Comments and Suggestions for Authors
Thank you for the opportunity to review your work. I have a few points to mention:
1. I am a bit confused about the rationale for targeting the VII trunk - what pathology are the authors hoping to address specifically? If it's to achieve balance for VII palsy, surely paralyzing a completely normal side wouldn't make sense? I think this is important to set in context the purpose of the study.
2. Introduction - if the authors want to mention US guided hydrodissection, then there needs to be a bit more explanation of the purpose of it, as it is not intuitive to the non-expert reader
3. Methods - it should be mentioned if the live participant was positioned similarly to the cadaver (to ensure consistent results)
4. Results
- in Figure 1, the legend includes abbreviations of a few structures but none of this was annotated in the figure itself?
- Figures 3, 6, 10 and 11 for the cadavers, some form of orientation grid (ie like a cross hair to show what is anterior, posterior, cranial, caudal etc.) would help.
- Line 142 mentions a distinct fascial boundary on sonographic imaging of the PBDM - can this be demonstrated on one of the Figures or a new Figure?
5. Discussion
- "previous ultrasound-based studies ... deeper anatomical context" should have accompanying reference(s)
- "this dual -pathway... nerve rehabilitation" should be expanded upon as I've mentioned in the beginning, to justify the rationale for the current study
- Hydrodissection potential - Were the cadaveric US-guided injections performed in a similar manner? This wasn't clear in the text. If not, this conclusion cannot be supported?
Author Response
Thank you for the opportunity to review your work. I have a few points to mention:
- - Thank you for your kind comment
Comment 1 I am a bit confused about the rationale for targeting the VII trunk - what pathology are the authors hoping to address specifically? If it's to achieve balance for VII palsy, surely paralyzing a completely normal side wouldn't make sense? I think this is important to set in context the purpose of the study.
- Response:We sincerely thank the reviewer for raising this crucial point. We agree that clarifying the clinical rationale is paramount. We have now revised both the Introduction and Discussion sections to explicitly state that the target for interventions like chemodenervation or neurolysis is the affected side in pathologies such as post-paralytic facial synkinesis or hemifacial spasm, not the healthy side. Furthermore, we have expanded the discussion to elaborate on other critical indications for FN trunk access, including diagnostic nerve blocks, neuromodulatory hydrodissection to potentially aid nerve recovery, and surgical planning for reanimation procedures. These additions, found in the second paragraph of the Discussion, provide the essential clinical context for our anatomical study.
Comment 2 Introduction - if the authors want to mention US guided hydrodissection, then there needs to be a bit more explanation of the purpose of it, as it is not intuitive to the non-expert reader
- Response:We thank the reviewer for this suggestion. As requested, we have significantly expanded the explanation of ultrasound-guided hydrodissection in the Introduction. The revised text now clearly outlines its dual purpose (mechanical decompression and biochemical neuroprotection), describes common injectates (D5W vs. local anesthetics/corticosteroids) and their proposed mechanisms, and explains its emerging role in motor nerve interventions. We believe this addition makes the technique accessible to a broader readership and firmly establishes the clinical relevance of our anatomical findings.
Comment 3 Methods - it should be mentioned if the live participant was positioned similarly to the cadaver (to ensure consistent results)
- Response:This is an excellent point regarding methodological consistency. We have added a clear statement in Section 2.6 (Live Doppler Ultrasound): "The participant was positioned in the same lateral decubitus position with induced lateral neck flexion as the cadaver to ensure anatomical consistency." This confirms that the sonographic landmarks identified in the cadaveric study were validated under identical positioning in a live subject, enhancing the reproducibility and reliability of our approach.
Comment 4 Results
- We appreciate the reviewer's detailed feedback on the figures.
in Figure 1, the legend includes abbreviations of a few structures but none of this was annotated in the figure itself?
- Response: Figure 1: We have replaced the original figure with an updated version where all key structures (FN, HN, VA, OA, PBDM, SP, MP, C1 TP) are now explicitly labeled with leader lines on both the schematic and the cadaveric image, eliminating any ambiguity.
Comment 5 Figures 3, 6, 10 and 11 for the cadavers, some form of orientation grid (ie like a cross hair to show what is anterior, posterior, cranial, caudal etc.) would help.
- Orientation Indicators: As suggested, we have added orientation markers (Anterior, Posterior, Superior, Inferior) to Figures 3, 6, 10, and 11 to provide readers with immediate anatomical context for the cadaveric dissection photos.
Comment 6 Line 142 mentions a distinct fascial boundary on sonographic imaging of the PBDM - can this be demonstrated on one of the Figures or a new Figure?
- Response:We thank the reviewer for this observation. The distinct fascial boundary is indeed visible in existing figures. We have now specifically referenced Figures 1, 4, and 7 in the main text (Section 3.3) where this sonographic feature can be clearly appreciated, guiding the reader to the relevant visual evidence.
Comment 7 Discussion "previous ultrasound-based studies ... deeper anatomical context" should have accompanying reference(s)
- Response: We have added supporting references (References 2,19) to the statement regarding the limitations of previous ultrasound studies, strengthening our claim.
Comment 8 "this dual -pathway... nerve rehabilitation" should be expanded upon as I've mentioned in the beginning, to justify the rationale for the current study
- Response:We have comprehensively expanded the paragraph on the dual-pathway approach in the Discussion. It now includes specific clinical examples (e.g., FN trunk block for synkinesis, hypoglossal nerve block for glossopharyngeal neuralgia) to powerfully justify the clinical need and rationale for our study, directly addressing the reviewer's initial concern.
Comment 9 Hydrodissection potential - Were the cadaveric US-guided injections performed in a similar manner? This wasn't clear in the text. If not, this conclusion cannot be supported?
- Response:This was a critical point. We have revised the Hydrodissection Potential subsection to clearly state that while the cadaveric injections used dye for validation, the needle trajectory and target are identical to those required for clinical hydrodissection. We now explicitly state that our study provides anatomic validation for the needle access route, which is a direct prerequisite for performing hydrodissection in patients.
Reviewer 3 Report
Comments and Suggestions for Authors
The article presents a novel and well-conceived ultrasound-guided approach to accessing the facial nerve trunk using anatomical and vascular landmarks. The integration of musculoskeletal and Doppler-guided vascular mapping represents a valuable advancement in minimally invasive facial nerve procedures. The anatomical rationale is sound, and the conclusion is well supported by the data.
However, the Methods section would benefit from additional detail, particularly regarding sample characteristics (e.g., age, sex, and condition of the cadaver), injection volume, and validation steps. The inclusion of quantitative outcome measures, even basic ones (e.g., accuracy of injection localization), would significantly enhance the scientific rigor. Furthermore, although the single cadaver and one live subject are appropriate for a feasibility study, this limitation should be more clearly acknowledged in the discussion.
No major issues with the language or formatting were found. Figures are appropriately labeled and support the text.
Author Response
Comment 1 The article presents a novel and well-conceived ultrasound-guided approach to accessing the facial nerve trunk using anatomical and vascular landmarks. The integration of musculoskeletal and Doppler-guided vascular mapping represents a valuable advancement in minimally invasive facial nerve procedures. The anatomical rationale is sound, and the conclusion is well supported by the data.
- We thank the reviewer for their positive assessment of our work.
Comment 2 However, the Methods section would benefit from additional detail, particularly regarding sample characteristics (e.g., age, sex, and condition of the cadaver), injection volume, and validation steps.
- Response: We thank the reviewer for pushing for greater methodological rigor. We have comprehensively updated the Methods section:
- Sample Characteristics : A non-embalmed fresh male cadaver (Case ID 25-078) was used for this feasibility study. The subject was 68.8 years old at the time of death, with a height of 166 cm and a body weight of 56 kg. The cause of death was lung cancer, and no prior surgical intervention in the head and neck region was identified. The cadaver was examined within 72 hours postmortem, thereby preserving near-physiological tissue elasticity and vascular integrity, which is essential for injection-based anatomical studies. This study was approved by the Institutional Review Board (IRB) of the Catholic University of Korea, College of Medicine (IRB No. MC24EISI0041).
- Injection Volume:The injection volume of 2 mL per site is now clearly stated in Section 2.4.
- Validation Steps:We have added an explicit description of the validation process to Section 2.5: "The distance between the dye and the facial nerve and hypoglossal nerve was measured to assess injection accuracy." This formally introduces the quantitative metric used.
Comment 3 The inclusion of quantitative outcome measures, even basic ones (e.g., accuracy of injection localization), would significantly enhance the scientific rigor.
- Response: We agree that quantitative data significantly strengthen the findings. We have now integrated the results of these measurements into Section 3.7, explicitly stating that the dye disseminated to within 2 mm of both the facial and hypoglossal nerves, providing a measurable outcome for the accuracy of our ultrasound-guided approach.
Comment 4 Furthermore, although the single cadaver and one live subject are appropriate for a feasibility study, this limitation should be more clearly acknowledged in the discussion.
- Response:We thank the reviewer for this critical point. We fully agree and have strengthened the Limitations subsection (4.2) to more explicitly and emphatically acknowledge this primary limitation. The opening sentence now clearly states: "This was an exploratory anatomical study using a single fresh-frozen cadaver, with vascular imaging supplemented by a live subject." Furthermore, we have added the following sentence to underscore the impact of this design: "The small sample size limits generalizability, and anatomical variation could influence reproducibility." We conclude the section by explicitly framing our findings as preliminary: "These findings should therefore be interpreted as preliminary guidance rather than definitive procedural standards." We believe these revisions provide an unambiguous and forthright discussion of the study's limitations.
Comment 5 No major issues with the language or formatting were found. Figures are appropriately labeled and support the text.
- Thank you for your kind comment
Round 2
Reviewer 2 Report
Comments and Suggestions for Authors
Thank you for addressing the points that have been raised. My only remaining issue is your argument about injecting the VII for post-paralytic facial synkinesis or hemifacial spasm. I think this is an over-simplification as the pathology behind synkinesis/hemifacial spasm is poorly understood to begin with, and I don't think injecting the VII trunk itself will address it adequately as there is some aberration in neural reinnervation following the original insult. In other words, treating the nerve trunk itself may not address the problem fully, and most certainly will knock out other facial muscles in the process, leading to further asymmetry. I think the best way forward is to remove the part on "post-paralytic facial synkinesis or hemifacial spasm" to avoid ambiguity. The rest of the changes are fine.
Author Response
Thank you for addressing the points that have been raised. My only remaining issue is your argument about injecting the VII for post-paralytic facial synkinesis or hemifacial spasm. I think this is an over-simplification as the pathology behind synkinesis/hemifacial spasm is poorly understood to begin with, and I don't think injecting the VII trunk itself will address it adequately as there is some aberration in neural reinnervation following the original insult. In other words, treating the nerve trunk itself may not address the problem fully, and most certainly will knock out other facial muscles in the process, leading to further asymmetry. I think the best way forward is to remove the part on "post-paralytic facial synkinesis or hemifacial spasm" to avoid ambiguity. The rest of the changes are fine.
Dear Reviewer,
Thank you for your final comment and for acknowledging that the rest of the changes are fine. We sincerely appreciate your expert insight regarding the pathophysiology of synkinesis and hemifacial spasm. We agree that targeting the trunk is an oversimplification of a complex neural rewiring process and may lead to unintended functional deficits.
As you suggested, we have removed the specific phrases "post-paralytic facial synkinesis or hemifacial spasm" from the Discussion section to avoid any ambiguity and overstatement of our technique's applications.
We believe this revision strengthens the manuscript by focusing on the anatomical feasibility of our approach while leaving its precise clinical applications to be defined by future, targeted clinical studies.
Thank you again for your guidance throughout the review process.
Sincerely,
The Authors